# Piperine Induces Apoptosis and Cell Cycle Arrest via Multiple Oxidative Stress Mechanisms and Regulation of PI3K/Akt and MAPK Signaling in Colorectal Cancer Cells

**DOI:** 10.3390/antiox14070892

**Published:** 2025-07-21

**Authors:** Wan-Ling Chang, Jyun-Yu Peng, Chain-Lang Hong, Pei-Ching Li, Soi Moi Chye, Fung-Jou Lu, Huei-Yu Lin, Ching-Hsein Chen

**Affiliations:** 1Department of Anesthesiology, Chang Gung Memorial Hospital at Chiayi, No. 8, West Section of Jiapu Road, Puzi City 613016, Chiayi County, Taiwan; chjack1975@yahoo.com.tw (W.-L.C.); pengjyun550@gmail.com (J.-Y.P.); leisure@cgmh.org.tw (C.-L.H.); peiching@cgmh.org.tw (P.-C.L.); 2School of Health Science, Division of Applied Biomedical Science and Biotechnology, IMU University, No. 126, Jalan Jalil Perkasa 19, Bukit Jalil, Kuala Lumpur 57000, Malaysia; chye_soimoi@imu.edu.my; 3Institute of Medicine, Chung Shan Medical University, No. 110, Section 1, Jianguo North Road, Taichung City 402306, Taiwan; fjlu@csmu.edu.tw; 4Department of Microbiology, Immunology and Biopharmaceuticals, College of Life Sciences, National Chiayi University, A25-303 Room, Life Sciences Hall, No. 300, Syuefu Road, Chiayi City 600355, Taiwan; abc25652652@gmail.com

**Keywords:** piperine, apoptosis, cell cycle arrest, oxidative stress, PI3K/Akt, MAPK, colorectal cancer

## Abstract

Piperine, a phytochemical alkaloid, exhibits notable anticancer properties in several cancer cell types. In this study, we investigated the mechanisms by which piperine induces cell death and apoptosis in colorectal cancer (CRC) cells, focusing on oxidative stress and key signaling pathways. Using MTT assay, flow cytometry, gene overexpression, and Western blot analysis, we observed that piperine significantly reduced cell viability, triggered G1 phase cell cycle arrest, and promoted apoptosis in DLD-1 cells. In addition, piperine effectively suppressed cell viability and induced apoptosis in other CRC cell lines, including SW480, HT-29, and Caco-2 cells. These effects were associated with increased intracellular reactive oxygen species (ROS) generation, mediated by the regulation of mitochondrial complex III, NADPH oxidase, and xanthine oxidase. Additionally, piperine modulated signaling pathways by inhibiting phosphoinositide 3-kinase (PI3K)/Akt, activating p38 and p-extracellular signal-regulated kinase (ERK). Pretreatment with antimycin A, apocynin, allopurinol, and PD98059, and the overexpression of p-Akt significantly recovered cell viability and reduced apoptosis, confirming the involvement of these pathways. This study is the first to demonstrate piperine induces apoptosis in CRC cells through a multifaceted oxidative stress mechanism and by critically modulating PI3K/Akt and ERK signaling pathways.

## 1. Introduction

Piperine is a phytochemical alkaloid that exhibits the potential to provide anticancer activities in CRC. In human HRT-18 rectal cancer cells, piperine-induced apoptosis and cytotoxicity are partially achieved through ROS [1]. Piperine stimulates autophagy-associated cell death by increasing ROS levels and blocking the Akt/mTOR pathway [2]. Additionally, piperine interferes with cell cycle progression, endoplasmic reticulum stress responses, and epithelial–mesenchymal transition through the modulation of GTPase signaling [3]. Other mechanisms include the suppression of the Wnt/β-catenin pathway in CRC cell lines to inhibit cell migration and proliferation [4].

Furthermore, piperine can be used in adjuvant radiotherapy and chemotherapy to enhance sensitivity to CRC. In combination with γ-radiation, it enhances G2/M phase arrest and apoptosis in HT-29 cells by increasing estrogen receptor beta expression [5]. It also acts as a chemosensitizer, enhancing the effects of 5-fluorouracil and celecoxib in chemoresistant CRC cells by modulating Wnt/β-catenin signaling [6,7]. These findings highlight piperine’s broad potential as both a direct and adjuvant anticancer agent.

ROS are highly reactive oxygen-containing species such as superoxide anion, hydrogen peroxide, and hydroxyl radicals. They are mainly generated within mitochondria but also arise from enzymatic sources including cyclooxygenase, xanthine oxidase, and NADPH oxidase [8,9,10,11,12]. Although low ROS levels support cellular proliferation, excessive ROS can induce apoptosis and inhibit tumor growth [8]. Many chemotherapeutic agents use this property by increasing intracellular ROS to trigger cancer cell death. In CRC, piperine has been observed to increase ROS and induce apoptosis, yet the precise molecular origins of this oxidative stress remain unclear.

Numerous anticancer agents induce ROS production to regulate the PI3K/Akt and MAPK signaling pathways, both of which are key regulators of cancer progression and therapy resistance. Compounds such as pyrvinium pamoate, antrodin C, metformin, lycorine, and delicaflavone have been reported to inhibit PI3K/Akt or activate MAPK pathways via ROS-mediated mechanisms [13,14,15,16,17]. Other natural compounds, including polyphyllin I, fibulin-5, quinalizarin, manumycin, and *Camellia ptilophylla* extracts, have also demonstrated ROS-dependent inhibition of proliferation through the modulation of these pathways [18,19,20,21,22]. These studies suggest a strong interconnection between oxidative stress and signal transduction in CRC.

Recent advances in CRC research emphasize the need for personalized therapeutic strategies, particularly for metastatic CRC. Targeted therapies directed at EGFR, VEGF, BRAF, and HER2, as well as immune checkpoint inhibitors (e.g., anti-PD-1), have demonstrated clinical benefit in biomarker-defined patient subgroups [23]. Molecular stratification based on RAS/BRAF mutation status, microsatellite instability, and tumor microenvironment profiling are now central to therapeutic decision-making [24,25]. However, drug resistance and tumor heterogeneity remain major obstacles, necessitating the exploration of novel agents that can modulate redox homeostasis and signaling pathways.

Given piperine’s ability to induce ROS and affect PI3K/Akt and MAPK signaling, it represents a promising candidate for integration into personalized CRC treatment strategies. In this study, we aim to elucidate the oxidative stress-related mechanisms by which piperine induces apoptosis in CRC cells. We focus on identifying the intracellular sources of ROS and evaluating the involvement of the PI3K/Akt and MAPK signaling pathways. These findings may contribute to the development of redox-targeted interventions and enhance the clinical utility of piperine in CRC therapy.

## 2. Materials and Methods

### 2.1. Reagents and Chemicals

Penicillin streptomycin–glutamine and fetal bovine serum (FBS) were bought from Gibco Inc. (Billings, MT, USA). The RPMI 1640 medium, Dulbecco’s Modified Eagle medium (DMEM), and Minimum Essential Medium (MEM) were obtained from Hyclone (South Logan, UT, USA). Piperine, trypan blue, crystal violet, DMSO, propidium iodide (PI), 2′,7′-dichlorodihydrofluorescein diacetate (DCFH-DA), MTT, Phosphate-buffered saline (PBS), nordihydroguaiaretic acid (Nordy), and other chemicals were bought from Sigma-Aldrich (St. Louis, MO, USA). The X-tremeGENE™ HP DNA transfection reagent was purchased from Roche (Raleigh, NC, USA). The protein assay kit was acquired from Bio-Rad Laboratories (Richmond, CA, USA). Primary antibodies against p53, Bax, p27, poly (ADP-ribose) polymerase (PARP), cyclin E, p-Akt, Akt, p-ERK, p-p38, p-JNK, tubulin, GAPDH and secondary antibodies were bought from Santa Cruz Biotechnology, Inc. (Santa Cruz, CA, USA). The 1036 pcDNA3 Myr HA Akt1 vector was obtained from the Addgene Company (Watertown, MA, USA). The pcDNA 3.1^(+)^ vector was bought from the Thermo Fisher Scientific Company (Taipei, Taiwan).

### 2.2. Cell Culture

The human CRC cell lines DLD-1, SW480, HT-29, and Caco-2 were used in this study. The DLD-1 and Caco-2 cell lines were purchased from the Bioresource Collection and Research Center, at the Food Industry Research and Development Institute (Hsinchu, Taiwan). The SW480 and HT-29 cell lines were obtained from the American Type Culture Collection (ATCC, Manassas, VA, USA). All cell lines are well-established human CRC models derived from patients with colorectal adenocarcinoma. Specifically, DLD-1 was derived from a 45-year-old male with Dukes’ C stage colorectal adenocarcinoma. SW480 originated from a primary colon adenocarcinoma of a 50-year-old male with Dukes’ B stage disease. HT-29 was isolated from a 44-year-old female with Dukes’ C stage, Grade II colorectal adenocarcinoma. Caco-2 was established from a 72-year-old male with well-differentiated colon adenocarcinoma.

The DLD-1 cells were cultured in an RPMI 1640 medium, the SW480 and HT-29 cells were cultured in a DMEM, and the Caco-2 cells were maintained in a MEM. All media were supplemented with 10% fetal bovine serum (FBS), 2 mM of L-glutamine, 100 units/mL of penicillin G, and 100 μg/mL of streptomycin. The cells were incubated at 37 °C in a humidified atmosphere containing 5% CO_2_. Stock solutions of piperine were prepared in DMSO, and the final concentrations were adjusted in the culture medium. The final DMSO concentration did not exceed 0.05%.

### 2.3. MTT Assay for Cell Viability

MTT assay was performed to determine cell viability. Approximately 4 × 10^4^ cells/well in 0.5 mL of an RPMI 1640 medium were incubated in 24-well plates. After growing overnight, the CRC cells were treated with piperine (0, 62.5, 125, and 250 μM), or various compounds for various time points. The plates were then added with 0.5 mg/mL of MTT solution and maintained at 37 °C for an additional 2 h. The supernatant was removed, and the formazan crystals were dispersed in 1 mL of DMSO. An aliquot of the DMSO-lysed solution (200 μL) was collected from every well and transferred to 96-well reader plates. Optical density was measured with a microplate reader (Bio-Rad, Richmond, CA, USA) at 570 nm.

### 2.4. Colony Formation Assay

DLD-1 cells were placed in 24-well culture plates (5 × 10^4^ cells/well) and cultured with piperine (0, 62.5, 125, and 250 μM) for 48 h. The colonies were then washed with PBS, fixed with a fixing solution (methanol–glacial acetic acid; 3:1) for 10 min, cleaned again with PBS, and stained with a 1% crystal violet solution for 30 min. Images were captured, and stained cell colony densities were determined using Image J software version 1.50d for analysis. To each well insert, 200 μL of the dissolving solution (33% acetic acid) was added to disperse the cells, and 100 μL of each dissolving solution was dispersed and measured at 570 nm by using a microplate reader (Bio-Rad, Richmond, CA, USA).

### 2.5. Cell Cycle Analysis

DLD-1 cells (5 × 10^5^) were incubated in six-well plates overnight and then cultured with piperine (0, 62.5, 125 and 250 μM) for 48 h. Cells were collected and fixed in a methanol solution (methanol–PBS = 2:1) at 4 °C for at least 24 h. After removing the methanol solution, the cells were stained with a PI solution (40 μg/mL) that contained DNase-free RNase A (40 μg/mL) for 30 min at room temperature in the dark and later assessed via CytoFLEX flow cytometry (Beckman Coulter, Brea, CA, USA) In the resulting histogram, the *X*-axis denotes PI fluorescence intensity while the *Y*-axis denotes the number of cells.

### 2.6. Terminal Deoxynucleotidyl Transferase dUTP Nick End Labeling (TUNEL) Analysis

Apoptosis was assessed via the TUNEL assay. After piperine incubation, DLD-1 cells were washed with PBS and fixed with 4% paraformaldehyde for 30 min at room temperature. The cells were cleaned with PBS and then added to a TUNEL reaction kit (BD Biosciences, San Jose, CA, USA) for 1 h at 37 °C. This procedure was followed by cleaning with PBS and a PI addition for 30 min. TUNEL-positive cells were distinguished through CytoFLEX flow cytometry (Beckman Coulter, Brea, CA, USA).

### 2.7. Western Blot

Proteins were separated using SDS–polyacrylamide gel electrophoresis and transferred to immobilon polyvinyldifluoride (PVDF) membranes (MilliporeSigma, Burlington, MA, USA). The PVDF membranes were blocked and probed using primary antibodies (1:1000) at 4 °C overnight. The blots were then incubated with secondary antibodies at room temperature for 1 h. The antigen–antibody complexes’ signals were evaluated via enhanced chemiluminescence (Amersham Pharmacia Biotech, Piscataway, NJ, USA) by using a luminescence image system (Hansor, New Taipei City, Taiwan).

### 2.8. Determination of Intracellular ROS

Intracellular ROS production was assessed via flow cytometry by using DCFH-DA staining after treating the DLD-1 cells (5 × 10^5^ cells/6 cm dish) with piperine (250 μM) for 1, 3, 6, and 24 h with 10 μM of DCFH-DA at 37 °C for 30 min in the dark and then stained with PI (4 μg/mL) to eliminate the dead cells. Intracellular ROS production was identified through the mean intensity of green fluorescence (2′,7′-DCF) via CytoFLEX flow cytometry (Beckman Coulter, Brea, CA, USA).

### 2.9. Overexpression of Akt in Cancer Cells

Approximately 6 μg of the 1036 pcDNA3 Myr HA Akt1 vector (Addgene, Cambridge, MA, USA) or the pcDNA 3.1^(+)^ vector (an empty vector; Thermo-Fisher Scientific, Taipei, Taiwan) was transfected to the DLD-1 cells via an X-treme transfection reagent. Approximately 4 × 10^4^ DLD-1 cells were cultured in a six-well plate and maintained in a 37 °C, 5% CO_2_ incubator for 24 h. Thereafter, 2 mL of a serum-free RPMI 1640 medium was substituted. Approximately 0.2 mL of the serum-free RPMI 1640 medium was added into an Eppendorf tube with 6 μL of X-tremeGENE HP DNA transfection reagent and 2 μg of vectors, gently pipetted, and mixed smoothly. After reacting at room temperature for 15 min, approximately 400 μL was collected and placed in a dish. The cells were mixed smoothly, returned to the incubator at 37 °C for 24 h, and then incubated with 1 μg/mL of puromycin antibiotics for 14 days to create a steady Akt overexpression cell line. The stable DLD-1 cell line was cultured in an RPMI 1640 medium with 10% FBS and then subjected to piperine for 48 h.

### 2.10. Statistical Analysis

Statistical analysis was performed using Student’s *t*-test with SigmaPlot 10.0 software. Data were presented as the mean ± standard deviation from at least three independent experiments. A *p* value of <0.05 was considered statistically significant. Due to the in vitro nature of this study, randomization and blinding were not applicable. Nonetheless, all procedures and reporting were conducted in accordance with the EQUATOR Network’s guiding principles for methodological transparency and reproducibility.

## 3. Results

### 3.1. Piperine Inhibits Cell Viability and Colony Formation in DLD-1 Cells

We examined the effects of piperine (0–250 μM) on the viability of DLD-1 cells over a 48 h period. Our results demonstrated that piperine significantly reduced cell viability in a concentration-dependent manner, with notable inhibition observed at 125 μM and 250 μM (Figure 1A). To further investigate the impact of piperine on cell proliferation, the DLD-1 cells were treated with increasing concentrations of piperine (0, 62.5, 125, and 250 μM) for 48 h, followed by colony formation assays. Piperine markedly suppressed colony-forming ability in a dose-dependent fashion (Figure 1B).

### 3.2. Piperine Induces Cell Cycle Arrest in DLD-1 Cells

To investigate whether the piperine-induced inhibition of colony formation was associated with cell cycle arrest, we performed a cell cycle analysis using flow cytometry. In the DLD-1 cells, piperine treatment for 48 h led to a concentration-dependent decrease in the S phase population (Figure 2A). Notably, treatment with 250 μM of piperine resulted in a significant accumulation of cells in the G1 phase, indicating G1 phase arrest. This observation was further supported by a Western blot analysis, which showed a decreased expression of cyclin E and an increased expression of p27 following piperine treatment (Figure 2B). These findings suggest that the antiproliferative effect of piperine on DLD-1 cells is mediated, at least in part, by the induction of G1 phase cell cycle arrest.

### 3.3. Piperine Induces Apoptosis in DLD-1 Cells

Next, we investigated the mode of cell death through which piperine exerts its cytotoxic effects on DLD-1 cells. To determine whether piperine induces apoptosis, DLD-1 cells were treated with piperine for 48 h, followed by a quantification of apoptotic cells using a TUNEL assay analyzed by flow cytometry. A dose-dependent increase in the proportion of TUNEL-positive cells was observed, indicating that piperine induces apoptosis in DLD-1 cells (Figure 3A). This observation was further supported by a Western blot analysis of apoptosis-related proteins. Piperine treatment led to an increased expression of cleaved PARP and Bax, further confirming the activation of apoptotic pathways (Figure 3B).

### 3.4. Piperine Induces ROS in DLD-1 Cells

To measure the probable participation of the intracellular ROS level of cells in the apoptotic effect of piperine, we determined intracellular ROS generation in the piperine-treated cells. Piperine treatment significantly increased intracellular ROS generation at 1 h and 3 h (Figure 4A). To identify the potential sources of piperine-induced ROS, DLD-1 cells were pretreated for 1 h with specific inhibitors targeting mitochondrial complexes I, II, and III, NADPH oxidase, lipoxygenase, and xanthine oxidase, followed by piperine treatment for 1 h. Intracellular ROS levels were then measured. Pretreatment with carboxin (a complex II inhibitor), antimycin A (a complex III inhibitor), Nordy (a lipoxygenase inhibitor), AEBSF and apocynin (NADPH oxidase inhibitors), and allopurinol (a xanthine oxidase inhibitor) significantly attenuated piperine-induced ROS production (Figure 4B). These results suggest that piperine stimulates ROS generation primarily through mitochondrial complexes II and III, lipoxygenase, NADPH oxidase, and xanthine oxidase pathways. To further clarify the individual effects of each inhibitor on basal ROS levels, we evaluated their actions in the absence of piperine (Figure 4C). Rotenone (a complex I inhibitor) alone markedly increased DCF fluorescence, indicating elevated ROS levels likely due to mitochondrial complex I inhibition. In contrast, carboxin, antimycin A, apocynin, AEBSF, allopurinol, and Nordy significantly reduced basal ROS levels compared to the untreated cells, indicating their suppressive effects on intrinsic ROS generation. These findings provide a reference framework for interpreting the combined effects of these inhibitors with piperine (Figure 4B).

### 3.5. Piperine-Mediated ROS Induces Cell Death and Apoptosis in DLD-1 Cells

To elucidate whether the sources of piperine-induced ROS contribute critically to DLD-1 cell death, cells were pretreated with specific inhibitors targeting mitochondrial complexes II and III, lipoxygenase, NADPH oxidase, and xanthine oxidase, followed by piperine exposure. Cell viability was subsequently assessed using the MTT assay. The piperine-induced reduction in cell viability was significantly attenuated by antimycin A (a mitochondrial complex III inhibitor), apocynin (an NADPH oxidase inhibitor), and allopurinol (a xanthine oxidase inhibitor) (Figure 5A). Consistently, the upregulation of cleaved PARP induced by piperine was markedly suppressed upon pretreatment with antimycin A, apocynin, and allopurinol (Figure 5B). These findings suggest that excessive ROS production originating from mitochondrial complex III, NADPH oxidase, and xanthine oxidase plays a pivotal role in mediating piperine-induced cytotoxicity and apoptosis in DLD-1 cells. In parallel, the individual cytotoxicity of these inhibitors was evaluated (Figure 5C). Rotenone and Nordy significantly reduced cell viability, consistent with their known mitochondrial and ferroptosis toxicity, respectively. However, most other inhibitors including apocynin, antimycin A, AEBSF, and allopurinol exhibited only mild effects on cell viability when used alone. These findings support the idea that the observed rescue effects in piperine co-treatment (Figure 5A) are likely due to pathway modulation rather than additive toxicity.

### 3.6. Piperine Inhibits p-Akt and Regulates MAPKs in DLD-1 Cells

Excessive ROS production is known to modulate the PI3K/Akt and MAPK signaling pathways, ultimately leading to apoptosis. To determine whether piperine affects these pathways, we examined the phosphorylation levels of Akt, p38, ERK, and JNK in piperine-treated cells using Western blot analysis. Piperine treatment markedly suppressed the phosphorylation of Akt in a concentration-dependent manner (Figure 6A). In contrast, piperine induced a significant increase in the phosphorylation of p38 and ERK (Figure 6B). Notably, the phosphorylation of JNK was reduced following piperine exposure (Figure 6B). These results suggest that piperine differentially modulates key signaling components of the PI3K/Akt and MAPK pathways, which may contribute to its pro-apoptotic effects in CRC cells.

### 3.7. Piperine Induces Apoptosis Through Akt and ERK Signaling Regulation

To further elucidate the involvement of the PI3K/Akt, p38, and ERK signaling pathways in piperine-induced cell death and apoptosis, we performed functional modulation experiments. DLD-1 cells were transfected with a p-Akt overexpression plasmid or pretreated with SB203580 (a p38 inhibitor) and PD98059 (an ERK inhibitor), followed by piperine treatment for 48 h. Cell viability was subsequently assessed. The overexpression of p-Akt, as well as pretreatment with SB203580 and PD98059, significantly restored cell viability compared to piperine treatment alone (Figure 7A,B). Furthermore, cleaved PARP levels were markedly reduced in cells with p-Akt overexpression (Figure 7C) and in those pretreated with PD98059 (Figure 7D). These findings indicate that the suppression of the PI3K/Akt pathway and the activation of the ERK pathway are key mechanisms underlying piperine-induced apoptosis and cytotoxicity in CRC cells.

### 3.8. Piperine Induces Cytotoxicity and Apoptosis in Multiple CRC Cell Lines

To assess the broader relevance of our findings, we examined piperine’s effects in three additional CRC cell lines: SW480, HT-29, and Caco-2. The MTT assay results showed a dose-dependent decrease in cell viability in all the tested cell lines after 48 h of treatment with piperine (Figure 8A). Moreover, a Western blot analysis demonstrated increased cleaved PARP levels, confirming apoptosis induction (Figure 8B). These results are consistent with our observations in DLD-1 cells, suggesting that piperine exerts its cytotoxic effects across different CRC models.

## 4. Discussion

ROS play pivotal roles in regulating cell death, proliferation, and differentiation through the modulation of various signaling molecules [26,27,28]. Compared to normal cells, cancer cells typically exhibit elevated basal ROS levels, which are closely associated with tumor initiation, metastasis, and drug resistance [29,30]. While moderate ROS levels may support cancer cell survival and progression, excessive ROS accumulation can surpass the cellular antioxidant defense capacity and induce cell death [31,32]. Therefore, modulating ROS levels represents a promising strategy for anticancer therapy. Several small molecules, such as furanodienone, have been reported to selectively induce ROS-mediated cytotoxicity in cancer cells [33]. In the present study, piperine treatment significantly elevated intracellular ROS levels at early time points (1 h and 3 h) (Figure 4A). Pretreatment with specific inhibitors, including carboxin (a mitochondrial complex II inhibitor), antimycin A (a mitochondrial complex III inhibitor), Nordy (a lipoxygenase inhibitor), AEBSF and apocynin (NADPH oxidase inhibitors), and allopurinol (a xanthine oxidase inhibitor), effectively attenuated piperine-induced ROS generation (Figure 4B). These findings suggest that piperine stimulates ROS production through multiple intracellular sources.

Our data suggest that rotenone, a known complex I inhibitor, exhibits synergistic effects when combined with piperine, which also appears to disrupt mitochondrial function. One possible explanation is that inhibition of complex I leads to electron accumulation and leakage, promoting superoxide formation through direct interaction with molecular oxygen. Additionally, blocking complex I may shift electron flow toward complexes II or III, further enhancing ROS production. Piperine may independently impair mitochondrial respiration, and in the presence of rotenone, these effects could be amplified, leading to elevated oxidative stress. Collectively, these findings support the notion that piperine-induced ROS generation is not solely mediated via complex I but may involve broader mitochondrial dysfunction and electron transport disruption. While the ability of antimycin A to increase ROS levels through complex III inhibition is well documented, our results demonstrated that pretreatment with antimycin A unexpectedly attenuated piperine-induced ROS generation. A similar effect was also observed with carboxin, a complex II inhibitor. This suggests that functional electron flow through mitochondrial complexes II and III may be essential for piperine to exert its full pro-oxidant effect. One plausible explanation is that piperine-induced ROS production relies on the accumulation and leakage of electrons during mitochondrial respiration. When electron flow is pharmacologically blocked at complex II or III, the disrupted transport may prevent electrons from reaching downstream sites where they would otherwise interact with molecular oxygen to generate ROS. This mechanistic interference likely explains the reduced ROS accumulation observed upon inhibitor co-treatment, even with agents like antimycin A that are otherwise pro-oxidant. These findings underscore that piperine-induced oxidative stress is not solely a result of random ROS upregulation but is mechanistically dependent on an intact and functioning electron transport chain. Among these ROS inhibitors, only antimycin A, apocynin, and allopurinol can recover cell viability (Figure 5A) and apoptosis (Figure 5B), indicating that ROS generated from mitochondrial complex III, NADPH oxidase, and xanthine oxidase are the major factors for piperine-induced cell death.

Under physiological conditions, the mitochondrial electron transport chain is responsible for generating approximately 90% of intracellular ROS [34]. During electron transport, a portion of electrons may prematurely leak to molecular oxygen, forming superoxide and other ROS [35]. Numerous natural compounds and phytochemicals exert anticancer effects by inducing mitochondrial dysfunction and enhancing ROS production in cancer cells. For example, azoxystrobin suppresses oral cancer progression by specifically inhibiting mitochondrial complex III activity, leading to ROS accumulation and apoptosis induction [36]. Similarly, ginsenoside Rh2, an active monomer isolated from ginseng, promotes mitochondrial ROS generation and triggers apoptosis in cervical cancer cells by targeting electron transfer chain complex III [37]. Taxodione and miltirone, both diterpenoid compounds, have also been reported to inhibit mitochondrial complexes III and V, resulting in elevated ROS levels and apoptosis in leukemia cells [38,39]. Furthermore, Kari et al. demonstrated that in glioblastoma multiforme, electron transport chain complex III can bypass complex I to elevate intracellular ROS levels through G protein-coupled receptor 17 signaling, offering novel therapeutic implications [40]. These findings collectively support our observation that pretreatment with antimycin A, a mitochondrial complex III inhibitor, significantly reversed the cytotoxic and pro-apoptotic effects of piperine in DLD-1 cells. This result highlights mitochondrial complex III as a critical source of piperine-induced oxidative stress and a potential mechanistic target in CRC therapy.

NADPH oxidase is recognized as a major endogenous source of ROS during tumorigenesis [41]. As stated earlier, the subunit Nox1 of NADPH oxidase is expressed in human colon cancer cells [42]. Various stimuli can activate NADPH oxidase in cancer cells [43]. At present, numerous studies have indicated that many anticancer compounds can affect NADPH oxidase activity, inducing intracellular ROS overproduction. For instance, lutein-induced ROS production in gastric cancer AGS cells is dependent on NADPH oxidase activity and is associated with apoptosis and NF-κB activation [44]. Imidazo [1,2-a] pyridine-based derivatives evidently stimulate cytotoxicity by notably augmenting NADPH oxidase activity, which leads to the stimulation of ROS-mediated apoptosis in A549 lung cancer cells [45]. Similarly, furanodienone induces apoptosis in CRC cells through ROS production originating from NADPH oxidase 4 [46], and glycyrrhetinic acid activates NADPH oxidases to promote ROS accumulation and ferroptosis in triple-negative breast cancer cells [47]. In line with these findings, our results demonstrated that pretreatment with apocynin, a NADPH oxidase inhibitor, significantly attenuated piperine-induced ROS generation (Figure 4B). Furthermore, apocynin pretreatment markedly restored cell viability (Figure 5A) and reduced the expression of cleaved PARP (Figure 5B) compared to piperine treatment alone. These results support the conclusion that piperine-mediated suppression of cell viability and its induction of apoptosis in DLD-1 cells are, at least in part, mediated through ROS generation via NADPH oxidase activation.

Although apocynin is widely used as a pharmacological inhibitor of NADPH oxidase, its classical inhibitory activity requires enzymatic activation by myeloperoxidase (MPO), an enzyme predominantly expressed in phagocytic immune cells. As DLD-1 CRC cells are non-phagocytic and unlikely to express high levels of MPO [48], the inhibitory effects of apocynin in this model may involve MPO-independent mechanisms or other redox-modulating properties. Therefore, the moderate yet statistically significant reduction in piperine-induced ROS following apocynin pretreatment should be interpreted with caution. Nonetheless, the consistent ROS-attenuating effects observed with both apocynin and AEBSF suggest that NADPH oxidase activity may partially contribute to piperine-induced oxidative stress in DLD-1 cells. In addition to apocynin, AEBSF was employed to further investigate the role of NADPH oxidase in piperine-induced ROS production. Although AEBSF is primarily recognized as a serine protease inhibitor, previous research by Diatchuk et al. demonstrated that AEBSF also inhibits NADPH oxidase activation by interfering with the assembly of its cytosolic and membrane-bound components [49]. Based on this dual functionality, AEBSF has been adopted as a pharmacological tool to explore NADPH oxidase involvement in ROS-related pathways. While its potential off-target effects cannot be excluded, the fact that AEBSF and apocynin exhibited consistent effects in attenuating piperine-induced ROS supports the possibility that NADPH oxidase contributes, at least in part, to the oxidative stress observed in DLD-1 cells.

Previous findings suggest a functional role of NOX1 in DLD-1 CRC cells. Cheng et al. [50] demonstrated that DLD-1 cells exhibit localized and regulated NOX1 activity mediated by p47^phox^-related organizer proteins. Additionally, Lu et al. [51] provided comprehensive expression data ranking CRC cell lines for NOX1 levels and documenting basal and PMA-induced superoxide production, further supporting the relevance of NOX1-mediated ROS generation in colorectal cancer. In our study, pretreatment with two NADPH oxidase inhibitors, apocynin and AEBSF, significantly attenuated piperine-induced ROS generation, supporting the contribution of NOX-dependent mechanisms. These results indicate that, although basal NOX expression may not be prominent, the enzyme system is pharmacologically responsive and functionally involved in oxidative signaling in DLD-1 cells. Future investigations using isoform-specific NOX inhibitors would help clarify the precise roles of individual NOX isoforms. Thus, considering that piperine’s anticancer efficiency may be partly attributed to its capacity to elevate intracellular ROS derived from an NADPH oxidase, initiating a sequence of mechanism cascade that leads to cellular apoptosis is reasonable.

Additionally, we examined the role of lipoxygenase using Nordy. Nordy pretreatment reduced piperine-induced ROS levels but unexpectedly enhanced piperine-induced cytotoxicity. This suggests that lipoxygenase inhibition may suppress stress-adaptive ROS responses or promote alternative, ROS-independent forms of cell death. Given the central role of lipoxygenase in ferroptosis, it is plausible that piperine may partially induce ferroptotic cell death. However, since ferroptosis-specific markers were not assessed in this study, further investigations using ferroptosis inhibitors and lipid peroxidation assays are needed to clarify this possibility.

Xanthine oxidase is a key metabolic enzyme involved in purine degradation and the oxidation of various xenobiotic substrates, generating ROS as a byproduct [52]. Under hypoxia or ischemia-reperfusion circumstances, xanthine oxidase activation has been associated with ROS-dependent tissue damage [53]. A previous study demonstrated that alternol, a fermented extract derived from the mutant fungus *Alternaria alternata* var. *monosporus*, functions as a unique xanthine dehydrogenase/xanthine oxidase activator in cancer cells, promoting ROS accumulation and subsequent apoptosis [11]. Moreover, xanthine oxidase has been shown to mediate the oxidative stress induced by various anticancer compounds; for instance, pretreatment with allopurinol, a selective xanthine oxidase inhibitor, effectively prevented berberine-induced oxidative stress in human prostate cancer PC-3 cells [54]. Consistent with these findings, our results revealed that pretreatment with allopurinol significantly restored cell viability (Figure 5A) and reduced cleaved PARP expression (Figure 5B) in piperine-treated DLD-1 cells. These observations suggest that xanthine oxidase contributes to piperine-induced ROS generation and plays a role in mediating its cytotoxic and pro-apoptotic effects in CRC cells.

The PI3K/Akt pathway plays an important role in drug resistance, metastasis, and cancer cell proliferation, and is widely recognized as a promising molecular target for cancer treatment [55]. Previous investigations have reported that elevated intracellular ROS levels can suppress the PI3K/Akt/mTOR signaling pathways, thereby inducing apoptosis in CRC cells [14,17,18,56]. For example, delicaflavone was shown to trigger ROS-mediated cell cycle arrest and apoptosis via the mitochondrial and endoplasmic reticulum stress pathways, in parallel with an inhibition of both the Ras/MEK/Erk and PI3K/Akt/mTOR signaling cascades [17]. Antrodin C, derived from *Antrodia cinnamomea*, promotes apoptosis through the activation of the ROS/Akt/ERK/p38 signaling pathway in CRC cells [14]. Similarly, Fibulin-5 induces apoptosis via the modulation of the Akt signal and ROS/MAPK pathways by downregulating the transient receptor potential cation channel subfamily V member 1 [18]. Wei-Tong-Xin has also been shown to promote the ROS-dependent, caspase-mediated apoptosis associated with PI3K/Akt pathway inhibition in CRC cells [56]. In the present study, Western blot analysis confirmed that piperine suppressed Akt phosphorylation in DLD-1 cells (Figure 6A). Moreover, an overexpression of constitutively active Akt partially restored piperine-reduced cell viability (Figure 7A) and diminished cleaved PARP expression compared to empty plasmid-transfected cells (Figure 7C). These findings indicate that the inhibition of the PI3K/Akt pathway contributes to piperine-induced cytotoxicity and apoptosis. However, given that Akt overexpression only partially rescued the cells from piperine-induced death, it is likely that additional pathways are also involved in mediating the overall cytotoxic effect of piperine.

The MAPK signaling pathway is downstream of ROS and plays a significant role in the stimulation of apoptosis [57]. Numerous studies have demonstrated that elevated ROS levels can induce cell death through the activation of MAPK pathways [58,59,60]. Among them, the p38 MAPK pathway is particularly sensitive to oxidative stress and is activated in response to increased intracellular ROS. The activation of the p38 pathway contributes to apoptosis induction, cell cycle arrest, and the inhibition of cellular proliferation [61,62,63]. Additionally, oxidative stress originating from the endoplasmic reticulum has been shown to stimulate MAPK signaling, particularly p38, which in turn mediates apoptotic responses or halts the cell cycle [64,65,66]. The phosphorylation of p38 is closely associated with increased ROS levels in cancer cells [67]. Traditionally, the ERK pathway, often activated by growth factors or oncogenic Ras, has been associated with promoting cell proliferation in cancer [68]. However, accumulating evidence suggests that ROS-dependent ERK activation can lead to apoptosis and cell cycle arrest in cancer cells [69]. For example, anticancer agents such as etoposide and cisplatin require sustained ERK activation to trigger apoptosis in several transformed or immortalized cells [70]. In our study, Western blot analysis revealed that piperine treatment significantly increased the phosphorylation levels of ERK and p38, while slightly reducing the phosphorylation of JNK (Figure 6B). Functionally, pretreatment with PD98059 (an ERK inhibitor) and SB203580 (a p38 inhibitor) partially reversed piperine-induced cell death in DLD-1 cells (Figure 7B). Notably, only PD98059 pretreatment effectively reduced cleaved PARP expression compared with piperine treatment alone (Figure 7D), indicating that the activation of the ERK pathway is a key signaling mechanism involved in piperine-induced cell death and apoptosis.

Our initial findings in the DLD-1 cells demonstrated that piperine induces apoptosis through the inhibition of the PI3K/Akt pathway and the activation of the ERK signaling pathway. To validate the generalizability of this mechanism, we extended our study to include three additional CRC cell lines, SW480, HT-29, and Caco-2, which were selected based on their known expression of p-Akt and p-ERK. These cell lines represent diverse clinical and genetic characteristics of CRC and have been reported to possess constitutively active Akt and ERK signaling, making them relevant models for this investigation [71,72,73,74]. Notably, a comprehensive molecular characterization of CRC cell lines has shown that HT-29 and DLD-1 can cluster into distinct molecular subgroups, highlighting that the models used in this study represent different intrinsic CRC subtypes rather than merely alternative cell lines [75]. Our data confirm that piperine treatment leads to a similar suppression of cell viability in these cell lines, accompanied by increased cleaved PARP expression. While published studies have primarily described ROS induction by piperine in CRC, there is limited literature specifically addressing piperine’s effects on the PI3K/Akt and MAPK pathways in non-DLD-1 models. Our data, although primarily mechanistically characterized in DLD-1 cells, provide novel supportive evidence that piperine’s oxidative stress-mediated apoptosis is not limited to a single CRC model.

DLD-1 cells are characterized by activating mutations in KRAS^G13D^ and PIK3CA^H1047R^, and a loss-of-function mutation in TP53, which collectively lead to the constitutive activation of oncogenic signaling pathways. These alterations could influence the magnitude or nature of piperine’s effects, particularly regarding PI3K/Akt and MAPK pathway regulation. However, the consistent effects observed in additional CRC cell lines with different mutational landscapes (e.g., SW480: KRAS^G12V^, HT-29: BRAF^V600E^, Caco-2: wild-type KRAS/TP53) suggest that the pro-apoptotic actions of piperine may not be strictly dependent on a specific mutation and are likely mediated through broader oxidative stress-related and mitochondrial mechanisms.

Although our findings demonstrated that piperine induces cell cycle arrest and apoptosis in CRC cells, the concentrations required (62.5–250 μM) may appear relatively high when compared to the typical pharmacologically active levels in vivo. Multiple studies have demonstrated that piperine exerts cytotoxic effects at concentrations ranging from 25 to 300 μM, including in CRC cell lines such as DLD-1, SW480, and HCT116, as well as in other cancer models [4,76]. These findings support the concentration range used in our in vitro experiments and underscore the mechanistic consistency of piperine-induced cell death and cell cycle arrest through ROS-mediated pathways and signaling modulation. Similar effective ranges have been reported in other studies on piperine’s anticancer activity [77,78]. It is important to note that piperine is a dietary alkaloid with inherently limited aqueous solubility and low oral bioavailability due to extensive first-pass metabolism and poor absorption in the gastrointestinal tract [79,80]. However, several studies have reported that nanoformulations, liposomes, and piperine derivatives can dramatically enhance its bioavailability and anticancer potency [81,82]. These delivery strategies not only improve systemic circulation and cellular uptake but also enable lower effective concentrations to be used. Additionally, piperine has been shown to act synergistically with conventional chemotherapeutics such as 5-fluorouracil, thereby potentially allowing dose reductions while enhancing efficacy [6]. Future research into formulation strategies and combination therapy will be essential for translating piperine into a clinically viable anticancer agent.

The current study reveals that piperine exerts anticancer effects in CRC cells through the induction of oxidative stress and the modulation of the PI3K/Akt and MAPK signaling pathways. These findings have meaningful clinical implications, as they suggest that piperine may serve as potential adjuvant agents in CRC therapy, particularly in tumors with elevated oxidative stress vulnerability or dysregulated redox signaling. The ability of piperine to target multiple ROS-generating systems, mitochondrial complex III, NADPH oxidase, and xanthine oxidase offers a broad-spectrum mechanism that may help overcome resistance to conventional chemotherapeutics. Furthermore, the suppression of PI3K/Akt and the activation of ERK pathways indicate that piperine may sensitize cancer cells to targeted pathway inhibitors or other pro-apoptotic agents. Future studies should validate these mechanisms in vivo using animal models or patient-derived CRC organoids and assess the pharmacokinetic behavior and safety profile of piperine in preclinical settings. These efforts may provide directions for developing piperine-based combinatory or personalized therapies in CRC treatment.

The mechanistic insights from this study provide a rationale for integrating piperine into targeted therapeutic strategies for CRC. By demonstrating that piperine suppresses the PI3K/Akt pathway and activates ERK signaling in ROS-dependent manners, our findings suggest that piperine could be particularly beneficial in CRC patients exhibiting an aberrant activation of these pathways. For instance, tumors harboring mutations in *PIK3CA* or upstream receptor tyrosine kinases may be sensitized to piperine-induced oxidative stress, potentially enhancing the efficacy of PI3K inhibitors or reducing resistance to such agents. Likewise, piperine’s modulation of mitochondrial and NADPH oxidase-derived ROS implies it may complement existing treatments by disrupting redox homeostasis in cancer cells, a vulnerability increasingly recognized in precision oncology. These properties may inform future stratified approaches in CRC management, where redox status and signaling pathway alterations are considered in therapy selection. Additional translational studies will be required to confirm these effects in patient-derived models and to define optimal therapeutic combinations and dosing regimens.

Despite the promising findings, several limitations of this study should be acknowledged. First, although we expanded the experimental design to include multiple CRC cell lines, our conclusions are still based solely on in vitro models. The absence of in vivo studies limits the direct translational relevance of our findings. Second, while we demonstrated that piperine induces ROS and modulates PI3K/Akt and MAPK signaling, the precise molecular targets upstream of these pathways remain to be elucidated. Third, we did not assess the pharmacokinetics or bioavailability of piperine at therapeutically effective concentrations in physiological systems, which is a critical factor for clinical application. Lastly, although ROS inhibitors and pathway modulators were used, genetic approaches such as siRNA knockdown would strengthen the causal interpretation of the mechanisms involved. Future studies incorporating animal models, patient-derived xenografts, and pharmacological profiling are warranted to validate the therapeutic potential of piperine in CRC.

## 5. Conclusions

Collectively, the present findings demonstrate that piperine is a promising natural compound with potential anticancer properties against CRC. Piperine effectively inhibited cell viability, induced G1 phase cell cycle arrest through the downregulation of cyclin E and upregulation of p27, and triggered apoptosis via multiple mechanisms. Notably, piperine-induced apoptosis was associated with excessive ROS generation originating from mitochondrial complex III, NADPH oxidase, and xanthine oxidase. In addition, piperine suppressed the PI3K/Akt signaling pathway while activating the ERK pathway, thereby contributing to its pro-apoptotic effects (Figure 9). Despite these encouraging results, further systematic preclinical investigations are required to substantiate its therapeutic potential and support future clinical applications.

## Figures and Tables

**Figure 1 antioxidants-14-00892-f001:**
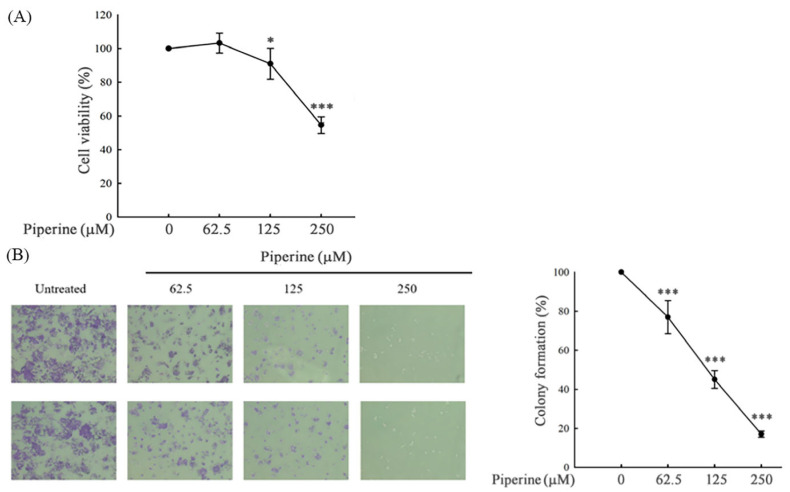
Piperine inhibits cell viability and colony formation in DLD-1 cells. (**A**) Cell viability of DLD-1 cells incubated for 48 h with different concentrations of piperine assessed via MTT assays. (**B**) Colony formation in DLD-1 cells incubated with piperine (0, 62.5, 125, and 250 μM) for 48 h (microscope magnification: 200×). Significant differences in the 0 μM-treated group are presented as *p* < 0.05 (*) and *p* < 0.001 (***).

**Figure 2 antioxidants-14-00892-f002:**
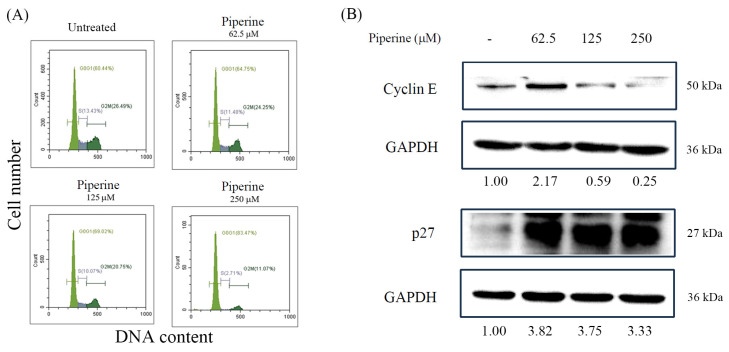
Piperine induces cell cycle G1 arrest in DLD-1 cells. (**A**) Cell cycle of DLD-1 cells cultured for 48 h with different concentrations of piperine assessed via PI staining and flow cytometry. (**B**) Expression of cell cycle-regulated proteins, cyclin E, and p27 in DLD-1 cells incubated with piperine (0, 62.5, 125, and 250 μM) for 48 h. The numbers below each band represent the relative densitometric values normalized to the untreated control (set as 1.00), and were calculated to illustrate the fold changes in protein expression.

**Figure 3 antioxidants-14-00892-f003:**
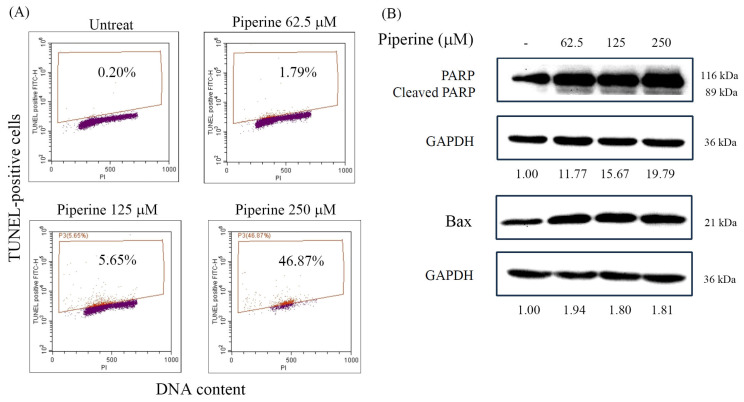
Piperine induces apoptosis in DLD-1 cells. (**A**) Percentages of apoptosis in DLD-1 cells incubated for 48 h with piperine (0, 62.5, 125, and 250 μM) assessed via TUNEL assay and flow cytometry. Boxes indicate TUNEL-positive cells, and the percentages shown within the boxes represent the proportion of TUNEL-positive cells. (**B**) Expression of apoptosis marker proteins, cleaved PARP, and Bax in DLD-1 cells incubated with piperine (0, 62.5, 125, and 250 μM) for 48 h. The numbers below each band represent the relative densitometric values normalized to the untreated control (set as 1.00), and were calculated to illustrate the fold changes in protein expression.

**Figure 4 antioxidants-14-00892-f004:**
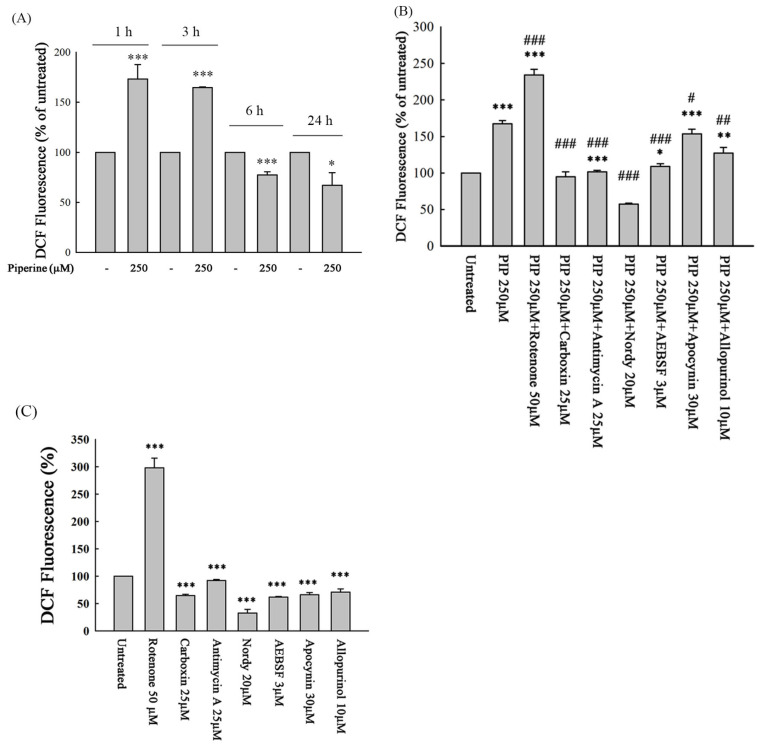
Piperine (PIP) induces ROS in DLD-1 cells. (**A**) Intracellular ROS levels in DLD-1 cells incubated with 250 μM of piperine for 1, 3, 6, and 24 h were assessed using DCFH-DA staining and flow cytometry. (**B**) Cells were pre-treated with various ROS-related inhibitors for 1 h followed by 250 μM piperine incubation for 1 h. Intracellular ROS levels were then analyzed using DCFH-DA staining and flow cytometry. (**C**) ROS levels in DLD-1 cells treated with each ROS-related inhibitor alone (without piperine) to assess their individual effects. ROS generation was evaluated by DCFH-DA staining and flow cytometry. Significant differences compared to the untreated group are indicated as * *p* < 0.05, ** *p* < 0.01, and *** *p* < 0.001. Significant differences compared to the piperine-treated group are indicated as # *p* < 0.05, ## *p* < 0.01, and ### *p* < 0.001.

**Figure 5 antioxidants-14-00892-f005:**
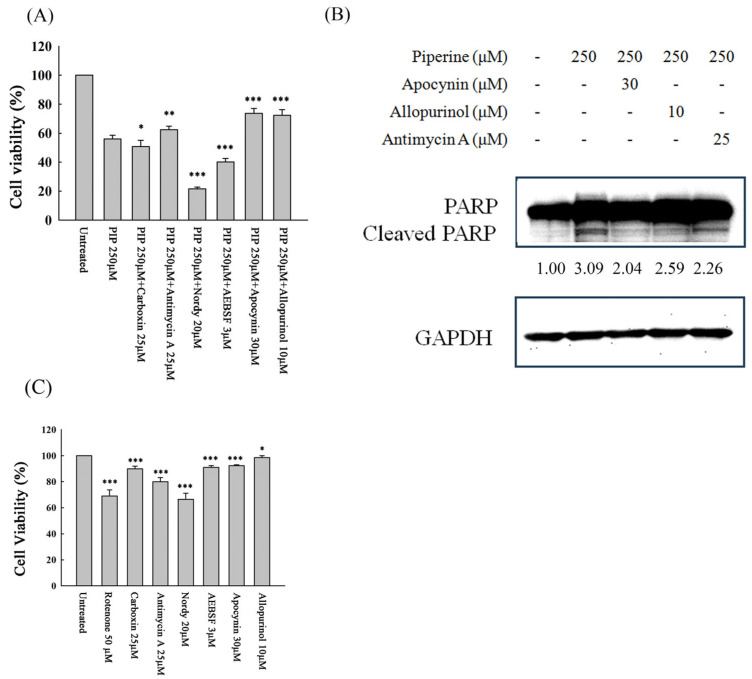
Piperine (PIP)-mediated ROS induces cell death and apoptosis in DLD-1 cells. (**A**) DLD-1 cells were pre-treated with various ROS-related inhibitors for 1 h, followed by 250 μM of piperine treatment for 48 h. Cell viability was assessed by MTT assay. Significant differences compared to the piperine-treated group are indicated as * *p* < 0.05, ** *p* < 0.01, and *** *p* < 0.001. (**B**) Cleaved PARP expression was analyzed by Western blot under the same treatment conditions as (**A**). (**C**) The cell viability of DLD-1 cells treated with each ROS-related inhibitor alone (without piperine) was evaluated using MTT assay to determine the individual cytotoxicity of each compound. Significant differences compared to the untreated group are indicated as * *p* < 0.05, and *** *p* < 0.001. The numbers below each band represent the relative densitometric values normalized to the untreated control (set as 1.00), and were calculated to illustrate the fold changes in protein expression.

**Figure 6 antioxidants-14-00892-f006:**
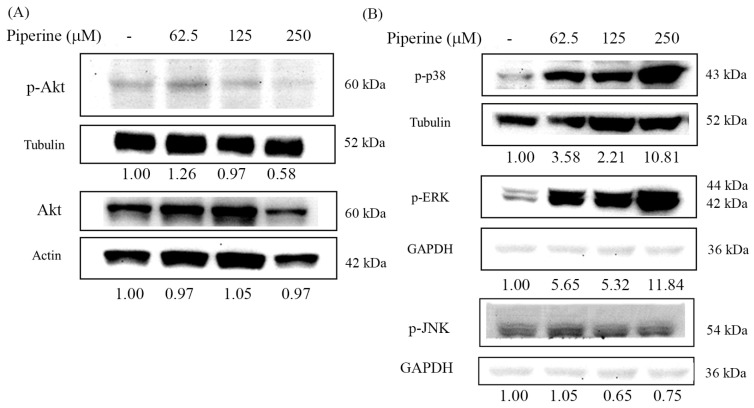
Piperine inhibits p-Akt and regulates MAPKs in DLD-1 cells. Expression of (**A**) p-Akt and Akt and (**B**) MAPK proteins in DLD-1 cells incubated with piperine (0, 62.5, 125, and 250 μM) for 48 h. GAPDH is shown as the same loading control for p-ERK and p-JNK because these proteins were run on the same gel and membrane. The numbers below each band represent the relative densitometric values normalized to the untreated control (set as 1.00), and were calculated to illustrate the fold changes in protein expression.

**Figure 7 antioxidants-14-00892-f007:**
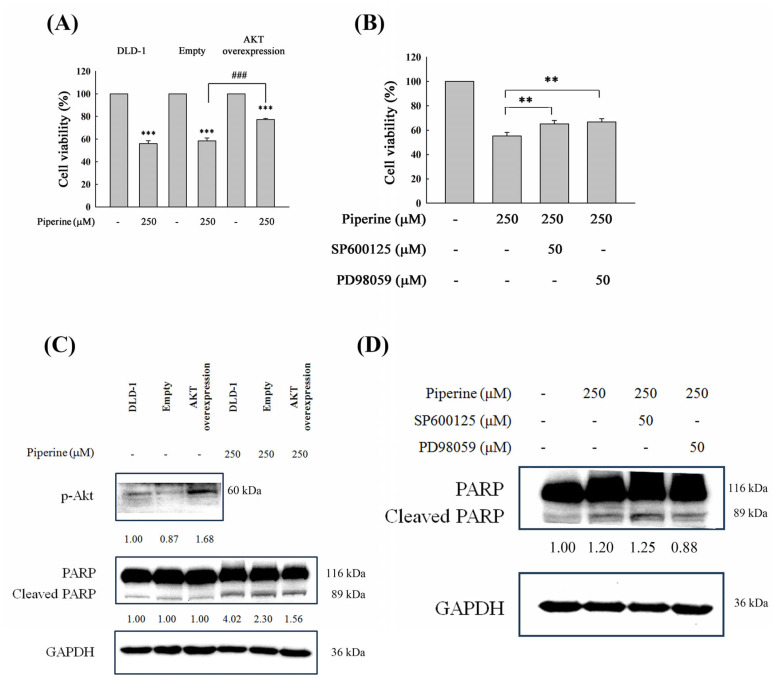
Piperine induces cell death and apoptosis through the Akt, p38, and ERK signaling pathways. (**A**) DLD-1 cells were transfected with empty plasmid or pAkt-overexpressed plasmid and then treated with 250 μM of piperine for 48 h. After treatment, cell viability was assessed via MTT assay. Significant differences in the untreated and piperine-treated empty plasmid groups are presented as *p* < 0.001 (***) and *p* < 0.001 (###), respectively. (**B**) Pretreatment of DLD-1 cells with 50 μM of SP600125 or 50 μM of PD98059 for 1 h, then incubation with 250 μM of piperine for 48 h. Cell viability was analyzed via MTT assay. Significant differences in the piperine-treated group are presented as *p* < 0.01 (**). (**C**) p-Akt expression was detected in the empty plasmid-, or p-Akt-overexpressed plasmid-transfected DLD-1 cells. Cells were incubated with 250 μM of piperine for 48 h. Cleaved PARP was measured via Western blot. (**D**) Pretreatment of DLD-1 cells with 50 μM of SP600125 or 50 μM of PD98059 for 1 h, then incubation with 250 μM of piperine for 48 h. Cleaved PARP was measured via Western blot. The numbers below each band represent the relative densitometric values normalized to the untreated control (set as 1.00), and were calculated to illustrate the fold changes in protein expression.

**Figure 8 antioxidants-14-00892-f008:**
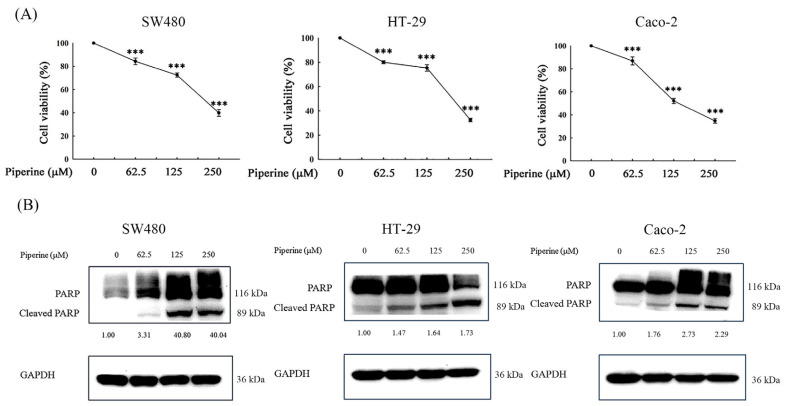
Piperine reduces cell viability and induces apoptosis in SW480, HT-29, and Caco-2 CRC cells. (**A**) Cell viability was assessed by MTT assay following treatment of SW480, HT-29, and Caco-2 cells with increasing concentrations of piperine (62.5–250 μM) for 48 h. Data are presented as mean ± SD from three independent experiments. *** *p* < 0.001 versus control. (**B**) Cleaved PARP protein expression was evaluated by Western blot under the same treatment conditions as (**A**) to confirm apoptosis induction. GAPDH served as the loading control. The numbers below each band represent the relative densitometric values normalized to the untreated control (set as 1.00), and were calculated to illustrate the fold changes in protein expression.

**Figure 9 antioxidants-14-00892-f009:**
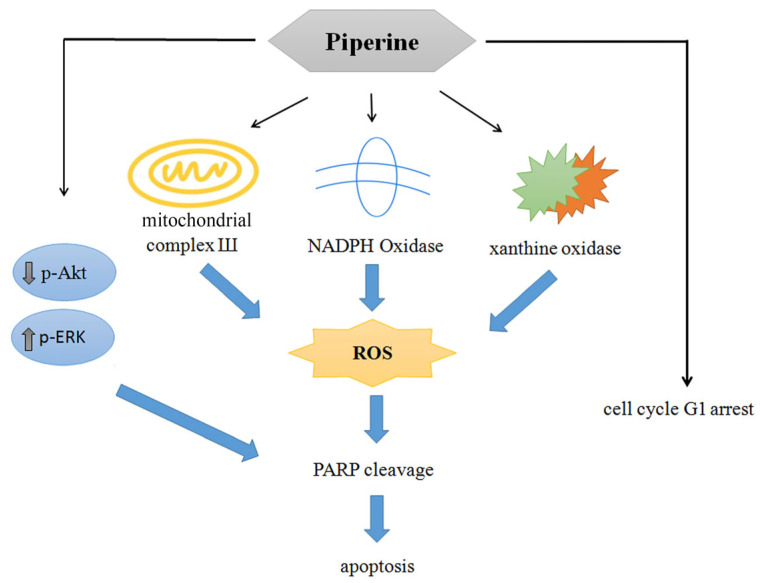
Proposed model of piperine-induced apoptosis in colorectal DLD-1 cancer cells.

## Data Availability

The data that support the findings of this study are contained within this article.

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
