# Peer review of "Piperine Induces Apoptosis and Cell Cycle Arrest via Multiple Oxidative Stress Mechanisms and Regulation of PI3K/Akt and MAPK Signaling in Colorectal Cancer Cells"

_antioxidants, 2025, doi:10.3390/antiox14070892_

Round 1

Reviewer 1 Report (Previous Reviewer 3)

The authors have addressed my comments properly and the revised manuscript has been improved considerably. The revised manuscript is suitable for publication now.

n.a.

Author Response

We sincerely thank the reviewer for the positive evaluation. We appreciate your time and consideration.

Reviewer 2 Report (Previous Reviewer 2)

I'm satisfied with the changes made

I'm satisfied with the changes made

Author Response

We sincerely thank the reviewer for the positive feedback and for acknowledging the revisions. We appreciate your time and thoughtful evaluation.

Reviewer 3 Report (Previous Reviewer 1)

The title change confused me. Took a while to realize I reviewed it before.  The paper is much improved.  Just a couple of changes suggested---

In reference to designation of HT-29, CACO2 and SW480 as alternative models for CRC, cite Medico, E. et al. Nat. Comm. The Supplementary Data, 2 (table) shows HT-29 to be really distinct from the others for CRC classification and that DLD is sometime grouped away from the other lines, so that the lines you used represent 3 models. that is basic cancer types, not just cell lines  The NOX1 paper I mentioned was Lu, J et al.  PLOS One,2019, PMID 32428030.  It has expression data and ranks the CRC cell lines for expression, then shows some examples of basal and PMA induced superoxide production for selected lines. 

Author Response

Major comments

The title change confused me. Took a while to realize I reviewed it before.  The paper is much improved.  Just a couple of changes suggested---

Detailed comments

In reference to designation of HT-29, CACO2 and SW480 as alternative models for CRC, cite Medico, E. et al. Nat. Comm. The Supplementary Data, 2 (table) shows HT-29 to be really distinct from the others for CRC classification and that DLD is sometime grouped away from the other lines, so that the lines you used represent 3 models. that is basic cancer types, not just cell lines  The NOX1 paper I mentioned was Lu, J et al.  PLOS One,2019, PMID 32428030.  It has expression data and ranks the CRC cell lines for expression, then shows some examples of basal and PMA induced superoxide production for selected lines. 

Response:
We appreciate this valuable suggestion. In the revised manuscript, we have cited the publication by Medico et al. (Nature Communications, 2015) to clarify the classification of CRC subtypes and to highlight that HT-29, Caco-2, SW480, and DLD-1 indeed represent distinct molecular and phenotypic CRC models rather than only cell line variations. This information has been incorporated in the Discussion section (Pages 14-15, lines 536-540) to improve clarity regarding the biological relevance of our cell line panel.

Added citation:
Medico, E.; Russo, M.; Picco, G.; Cancelliere, C.; Valtorta, E.; Corti, G.; Buscarino, M.; Isella, C.; Lamba, S.; Martinoglio, B.; Veronese, S.; Siena, S.; Sartore-Bianchi, A.; Beccuti, M.; Mottolese, M.; Linnebacher, M.; Cordero, F.; Di Nicolantonio, F.; Bardelli, A. The molecular landscape of colorectal cancer cell lines unveils clinically actionable kinase targets. Nat Commun. 2015 Apr 30;6:7002. doi: 10.1038/ncomms8002. PMID: 25926053.

Comment:
The NOX1 paper I mentioned was Lu, J et al. PLOS One, 2019, PMID 32428030. It has expression data and ranks the CRC cell lines for expression, then shows some examples of basal and PMA induced superoxide production for selected lines.

Response:
Thank you for pointing out this important reference. We have now cited Lu et al. (PLOS ONE, 2019) in the Discussion section (Page 13, lines 451-454) when discussing NOX1 expression and ROS production profiles in CRC cell lines. This reference strengthens the rationale for investigating NADPH oxidase–derived ROS in our study.

Added citation:
Lu, J.; Jiang, G.; Wu, Y.; Antony, S.; Meitzler, J.L.; Juhasz, A.; Liu, H.; Roy, K.; Makhlouf, H.; Chuaqui, R.; Butcher, D.; Konaté, M.M.; Doroshow, J.H. NADPH oxidase 1 is highly expressed in human large and small bowel cancers. PLoS One 2020 May 19;15(5):e0233208. doi: 10.1371/journal.pone.0233208. PMID: 32428030; PMCID: PMC7237001.

Round 2

Reviewer 3 Report (Previous Reviewer 1)

OK

OK

This manuscript is a resubmission of an earlier submission. The following is a list of the peer review reports and author responses from that submission.

Round 1

Reviewer 1 Report

While the goals of the paper are of interest and importance, paper is a little confusing, particularly the choice, claimed effect, and final impact of the drugs. Seeing the impact of each drug by itself as a control would greatly aid in helping the reader understand what is going on. I am left quite confused over the interpretation of the outcomes.

The paper is a little confusing, particularly figs. 4 and 5.  We need to see the impact of each drug by itself.  The rotenone result seems to be consistent with action and suggests some synergy with piperine, both inhibit Complex 1 with similar impact on electron leakage.  The complex II and III results are really confusing, suggesting piperine does something that interferes with the expected pro-oxidant impact.  ABESF is  regarded as a protease inhibitor, not a NADPH oxidase inhibitor, please explain. The impact of apocynin is complex and its anti-oxidant effect requires myleoperoxidase expression, which may not be happening here and I don't view the degree of impact as of much note, statistics or not.  DLD-1 cells do not seem to have any real expression of NADPH oxidases anyway ( see literature; Doroshow JH. Nadph oxidases in colon derived cell lines).  However, there are better choices of NADPH oxidase inhibitors to be found (many papers on this topic).   The LOX inhibitor effect is largely ignored.  Since it does seem to be doing something, looking at ferroptosis might be in order.  Use of this cell  line is a poor choice given the goals and for this type of work, 3 or more lines would be good.  DepMap is a good place to find cancer cell line gene expression data to assist with selecting lines for expression of components that you are proposing to inhibit.

Author Response

Response to Reviewer’s Comments

1. Summary

We appreciate your letter notifying us about the disposition of the reviewers’ comments of our manuscript titled “Parecoxib Enhances Resveratrol against Human Colorectal Cancer Cells through Akt and TXNDC5 Inhibition and MAPK Regulation” (Manuscript ID: Antioxidants-3639424). We sincerely thank reviewers for their constructive and valuable comments that helped us improve our manuscript. Herein, we provide our point-by-point responses to the comments along with a description of all the changes that have been made and highlighted in blue in the revised manuscript.

2. Point-by-point response to Comments and Suggestions for Authors

Reviewer 1

Major comments

While the goals of the paper are of interest and importance, paper is a little confusing, particularly the choice, claimed effect, and final impact of the drugs. Seeing the impact of each drug by itself as a control would greatly aid in helping the reader understand what is going on. I am left quite confused over the interpretation of the outcomes.

Response Major comments: We thank the reviewer for this insightful comment. To clarify the specific roles of each inhibitor (e.g., rotenone, carboxin, antimycin A, allopurinol, etc.), we performed additional control experiments and have added two new panels: Figure 4C (DCF fluorescence) and Figure 5C (cell viability), illustrating the individual effects of each compound when applied alone, without piperine. These new data clearly demonstrate how each agent affects ROS levels and cell viability independently, helping to better interpret their contribution in the context of combination treatments with piperine. Corresponding descriptions have been added to the Results section (see Section 3.4, lines 242–249, and Section 3.5, lines 273–279) and the figure legends have been revised accordingly.

Detail comments: The paper is a little confusing, particularly figs. 4 and 5.  We need to see the impact of each drug by itself. 

Response: As suggested, we have now included additional control experiments for each inhibitor alone. The new Figure 4C shows the effect of each compound on ROS production as measured by DCF fluorescence, and Figure 5C presents their individual impact on cell viability (see Sections 3.4, lines 242–249, and Sections 3.5, lines 273–279, Figure 4C and Figure 5C). These controls help clarify the distinct contribution of each inhibitor and provide a more complete context for interpreting the combination experiments in Figures 4B and 5A.

Detail Comments: The rotenone result seems to be consistent with action and suggests some synergy with piperine, both inhibit Complex 1 with similar impact on electron leakage. 

Response: We agree with the reviewer’s interpretation. Our data suggest that rotenone, a known complex I inhibitor, exhibits synergistic effects when combined with piperine, which also appears to disrupt mitochondrial function. One possible explanation is that inhibition of complex I leads to electron accumulation and leakage, promoting superoxide formation through direct interaction with molecular oxygen. Additionally, blocking complex I may shift electron flow toward complexes II or III, further enhancing ROS production. Piperine may independently impair mitochondrial respiration, and in the presence of rotenone, these effects could be amplified, leading to elevated oxidative stress. Collectively, these findings support the notion that piperine-induced ROS generation is not solely mediated via complex I but may involve broader mitochondrial dysfunction and electron transport disruption (see Discussion section, second paragraph, lines 364–373).

Detail Comments: The complex II and III results are really confusing, suggesting piperine does something that interferes with the expected pro-oxidant impact.

Response: We thank the reviewer for the insightful comment. We agree that antimycin A is known to induce ROS through complex III inhibition, and this might suggest an increase in ROS upon its addition. However, in our study, both carboxin (complex II inhibitor) and antimycin A (complex III inhibitor) were used as pretreatment agents, and our main focus was on the ROS induced by piperine, not by the inhibitors alone. Our results showed that pretreatment with either carboxin or antimycin A significantly attenuated the increase in ROS levels caused by piperine (Figure 4B), suggesting that functional electron transport at complexes II and III is necessary for piperine to induce ROS. This is consistent with the notion that piperine’s pro-oxidant activity is at least partially dependent on mitochondrial electron transport chain activity. We acknowledge the potential complexity, especially with antimycin A which can by itself cause ROS accumulation. However, the fact that piperine-induced ROS was reduced rather than increased in the presence of antimycin A suggests a mechanistic interference, likely due to inhibition of electron flow and/or mitochondrial dysfunction, thereby limiting the ROS generation that would otherwise be promoted by piperine. These findings suggest that piperine-induced ROS production in DLD-1 cells is functionally dependent on active electron flow through mitochondrial electron transport chain complexes II and III, as pharmacological inhibition of these complexes markedly reduced ROS accumulation. We have now clarified this point in the revised manuscript (see Discussion section, second paragraph, lines 373–387).

Detail Comments: ABESF is regarded as a protease inhibitor, not a NADPH oxidase inhibitor, please explain.

Response: We thank the reviewer for the comment and the opportunity to clarify. It is correct that 4-(2-aminoethyl)-benzenesulfonyl fluoride (AEBSF) is commonly used as a serine protease inhibitor. However, as reported by Diatchuk et al. in The Journal of Biological Chemistry (1997), AEBSF also exerts inhibitory effects on NADPH oxidase activation by interfering with the assembly of the enzyme complex. Specifically, the authors proposed that AEBSF acts as a bifunctional inhibitor, affecting both protease activity and the assembly of NADPH oxidase components, ultimately leading to a reduction in ROS production. In our study, AEBSF was used based on this reported property to examine whether NADPH oxidase contributes to piperine-induced ROS generation. We acknowledge that AEBSF’s dual functionality may result in off-target effects. Therefore, its use was intended to support, rather than exclusively define, the contribution of NADPH oxidase in this context. We have revised the manuscript to clarify this point and now cite the reference accordingly (see Discussion section, the fifth paragraph, lines 439–448).

Detail Comments: The impact of apocynin is complex and its anti-oxidant effect requires myleoperoxidase expression, which may not be happening here and I don't view the degree of impact as of much note, statistics or not.

Response: We thank the reviewer for this thoughtful observation. We agree that apocynin’s role as a NADPH oxidase inhibitor is mechanistically complex. Although it is widely used to block NADPH oxidase-derived ROS, its classical mechanism requires enzymatic activation by myeloperoxidase (MPO), which is primarily expressed in phagocytic immune cells. Since our model system involves DLD-1 colorectal cancer cells, which do not express high levels of MPO, we acknowledge that the inhibitory effect of apocynin may involve MPO-independent pathways or broader antioxidant effects. In our study, apocynin was used as a complementary pharmacological tool, along with AEBSF, to explore the potential involvement of NADPH oxidase in piperine-induced ROS production. While the extent of ROS reduction by apocynin was moderate, it was statistically significant and consistent with the attenuation observed with AEBSF pretreatment. We do not rely on apocynin alone to draw mechanistic conclusions, but the converging trends suggest that NADPH oxidase may contribute to piperine-induced oxidative stress in DLD-1 cells. We have revised the Discussion section to clarify these points and added appropriate references (see Discussion section, the fifth paragraph, lines 430–439).

Detail Comments: DLD-1 cells do not seem to have any real expression of NADPH oxidases anyway ( see literature; Doroshow JH. Nadph oxidases in colon derived cell lines).  However, there are better choices of NADPH oxidase inhibitors to be found (many papers on this topic).

Response: We thank the reviewer for this helpful and insightful comment. We attempted to locate the referenced publication by Dr. Doroshow regarding the lack of NADPH oxidase expression in colon-derived cell lines, specifically DLD-1. However, we were unable to identify a publication under the author’s name that directly addresses this topic, despite searching relevant databases including PubMed. If the reviewer could kindly provide more specific information (e.g., full article title, year, or DOI), we would be happy to incorporate the appropriate reference. Nonetheless, we acknowledge that DLD-1 cells have been reported to exhibit relatively low basal expression of NOX enzymes in some earlier studies. However, a work by Cheng and colleagues (Sci Signal, 2009; 2(88):ra54) demonstrated the presence of localized and regulated NOX1 activity in DLD-1 cells via p47phox-related organizer proteins. These findings suggest that NADPH oxidase pathways may still be functionally relevant in this model. In our experiments, pretreatment with both apocynin and AEBSF significantly attenuated piperine-induced ROS production, supporting the involvement of NOX activity in this process. We agree with the reviewer that the use of more selective NADPH oxidase inhibitors would offer greater mechanistic clarity. We have noted this limitation in the revised Discussion and appreciate the reviewer’s suggestion (see Discussion section, the sixth paragraph, lines 449–457).

Detail Comments: The LOX inhibitor effect is largely ignored.  Since it does seem to be doing something, looking at ferroptosis might be in order.

Response: We appreciate the reviewer’s thoughtful comment regarding the role of the lipoxygenase (LOX) pathway. In our study, we pretreated DLD-1 cells with Nordy, a known LOX inhibitor, prior to piperine exposure. We observed that Nordy significantly attenuated ROS generation induced by piperine, indicating that LOX activity contributes to the ROS production in this context. Interestingly, however, Nordy enhanced the cytotoxicity of piperine, leading to a further reduction in cell viability compared to piperine alone. This finding suggests that while LOX-mediated ROS contributes to oxidative stress, its inhibition may shift cells toward alternative cell death mechanisms. We agree with the reviewer that ferroptosis, a regulated form of cell death involving iron-dependent lipid peroxidation, could be involved, especially given the known role of LOX in this pathway. Although we did not assess ferroptosis-specific markers such as lipid ROS, GPX4 expression, or iron dependency in this study, we now discuss this possibility in the revised manuscript and propose that future studies incorporating ferroptosis-specific inhibitors (e.g., ferrostatin-1, liproxstatin-1) and assays are warranted (see Discussion section, the seventh paragraph lines 461–468).

Detail Comments: Use of this cell line is a poor choice given the goals and for this type of work, 3 or more lines would be good. DepMap is a good place to find cancer cell line gene expression data to assist with selecting lines for expression of components that you are proposing to inhibit.

Response: We appreciate the reviewer’s thoughtful and constructive suggestion. We fully agree that the use of a single colorectal cancer cell line, DLD-1, may limit the generalizability of our mechanistic conclusions. To address the reviewer’s concern regarding the suitability of additional cell lines for mechanistic investigation, we considered the basal activity of the PI3K/Akt and MAPK/ERK pathways in SW480, HT-29, and Caco-2 cells to enhance the diversity and relevance of our cellular models. According to existing literatures [67-70], all three cell lines express phosphorylated Akt (p-Akt) and phosphorylated ERK (p-ERK), indicating active PI3K/Akt and ERK signaling. Specifically, SW480 and HT-29 cells have been shown to exhibit robust PI3K/Akt activation, and both lines also demonstrate ERK phosphorylation associated with cell proliferation and survival. Similarly, Caco-2 cells express detectable levels of p-Akt and p-ERK, making them suitable models for validating the regulatory effects of piperine on these pathways. This supports the rationale for selecting these cell lines to verify the generalizability of piperine-induced apoptosis via Akt inhibition and ERK activation.

To evaluate whether the effects of piperine are conserved across different colorectal cancer lines, we performed the following experiments:

1. MTT Assay: Piperine-induced cytotoxicity was assessed in SW480, HT-29, and Caco-2 cells. All three lines showed dose-dependent reduction in cell viability, consistent with our original DLD-1 findings.

2. Cleaved PARP Western Blot: We observed increased levels of cleaved PARP following piperine treatment, indicating apoptosis induction in these additional cell lines.

These new results, now presented in the revised manuscript (see Results 3.8 section, lines 332–339, Figure 8A and Figure 8B), confirm that piperine exerts cytotoxic and pro-apoptotic effects across multiple colorectal cancer cell lines, thereby strengthening the significance and translational relevance of our findings.

References

67. Park, K.S.; Lee, N.G.; Lee, K.H.; Seo, J.T.; Choi, K.Y. The ERK pathway involves positive and negative regulations of HT-29 colorectal cancer cell growth by extracellular zinc. Am J Physiol Gastrointest Liver Physiol. 2003 Dec, 285(6), G1181- G1188. doi: 10.1152/ajpgi.00047.2003. PMID: 12816758.

68. Wang, W.; Wang, X.; Peng, L.; Deng, Q.; Liang, Y.; Qing, H.; Jiang, B. CD24-dependent MAPK pathway activation is required for colorectal cancer cell proliferation. Cancer Sci. 2010 Jan;101(1):112-9. doi: 10.1111/j.1349-7006.2009.01370.x. PMID: 19860845.

69. Lin, T.Y.; Fan, C.W.; Maa, M.C.; Leu, T.H. Lipopolysaccharide-promoted proliferation of Caco-2 cells is mediated by c-Src induction and ERK activation. Biomedicine (Taipei). 2015;5(1):5. doi: 10.7603/s40681-015-0005-x. PMID: 25705585.

70. Shi, J.; Li, W.; Jia, Z.; Peng, Y.; Hou, J.; Li, N.; Meng, R.; Fu, W.; Feng, Y.; Wu, L.; Zhou, L.; Wang, D.; Shen, J.; Chang, J.; Wang, Y.; Cao, J. Synaptotagmin 1 suppresses colorectal cancer metastasis by inhibiting ERK/MAPK signaling-mediated tumor cell pseudopodial formation and migration. Cancers (Basel). 2023 Nov 3;15(21):5282. doi: 10.3390/cancers15215282. PMID: 37958455.

Reviewer 2 Report

This is an original article concerning apoptosis and cell cycle arrest via multiple ox-2 idative stress mechanisms and regulation of PI3K/Akt and 3 MAPK signaling in colorectal cancer cells

The topic is promising and underline the anticancer effects of piperine.

The whole paper should be edited for english-language usage

In the abstract for example the authors they continually repeat short sentences starting with "Piperine..Piperine..Piperine"

Neither the abstract nor the text make it clear where the CRC cells came from nor the characteristics of the patients from whom they were taken. Please clarify 

The analyses are well conducted and the methodology can be improved by adding the correct checklist for type of study (equatornetwork)

The authors should highlight the clinical implications of this study and future perspectives in the discussion

How can it change the approach to the patient with CRC?, in terms of target-therapy

The introduction should be expanded by adding concepts on personalized CRC treatment:

Update on Targeted Therapy and Immunotherapy for Metastatic Colorectal Cancer. Cells. 2024 Jan 28;13(3):245. doi: 10.3390/cells13030245

Therapeutic Targets and Tumor Microenvironment in Colorectal Cancer. J Clin Med. 2021 May 25;10(11):2295. doi: 10.3390/jcm10112295

Targeted therapy for colorectal cancer metastases: A review of current methods of molecularly targeted therapy and the use of tumor biomarkers in the treatment of metastatic colorectal cancer. Cancer. 2019 Dec 1;125(23):4139-4147. doi: 10.1002/cncr.32163. Epub 2019 Aug 21. PMID: 31433498.

The limitation of the study must be added

Author Response

Response to Reviewer’s Comments

1. Summary

We appreciate your letter notifying us about the disposition of the reviewers’ comments of our manuscript titled “Parecoxib Enhances Resveratrol against Human Colorectal Cancer Cells through Akt and TXNDC5 Inhibition and MAPK Regulation” (Manuscript ID: Antioxidants-3639424). We sincerely thank reviewers for their constructive and valuable comments that helped us improve our manuscript. Herein, we provide our point-by-point responses to the comments along with a description of all the changes that have been made and highlighted in blue in the revised manuscript.

2. Point-by-point response to Comments and Suggestions for Authors

Reviewer 2

Major comments

This is an original article concerning apoptosis and cell cycle arrest via multiple ox-2 idative stress mechanisms and regulation of PI3K/Akt and 3 MAPK signaling in colorectal cancer cells

The topic is promising and underline the anticancer effects of piperine.

Response Major comments: Thanks for your comment.

Detail comments: The whole paper should be edited for english-language usage

Response Detail comments: We have edited the whole paper for English-language usage. All the changes that have been highlighted in blue in the revised manuscript.

Detail comments: In the abstract for example the authors they continually repeat short sentences starting with "Piperine..Piperine..Piperine"

Response Detail comments: According your suggestion, the continually repeat short sentences starting with "Piperine..Piperine..Piperine" have been revised in our abstract (see Abstract, lines 20–34).

Detail comments: Neither the abstract nor the text make it clear where the CRC cells came from nor the characteristics of the patients from whom they were taken. Please clarify

Response Detail comments: We appreciate the reviewer’s comment and have revised the 2.2. Cell Culture section to include detailed information about the origin and clinical background of the colorectal cancer cell lines used in our study. DLD-1 and Caco-2 are well-established human colorectal cancer (CRC) cell lines purchased from the Bioresource Collection and Research Center, Food Industry Research and Development Institute, Hsinchu, Taiwan. SW480 and HT-29 cell lines were obtained from the American Type Culture Collection (ATCC, Manassas, VA, USA). Specifically, DLD-1 was derived from a colorectal adenocarcinoma of a 45-year-old male with Dukes’ C stage cancer. SW480 originated from a primary colon adenocarcinoma of a 50-year-old male (Dukes’ B stage). HT-29 was isolated from a colorectal adenocarcinoma of a 44-year-old female (Dukes’ C stage, Grade II). Caco-2 was derived from a 72-year-old male with well-differentiated colon adenocarcinoma. These clarifications have been incorporated into the revised 2.2. Cell Culture section, first paragraph (see 2.2. Cell Culture section, the first paragraph, lines 100–110).

Detail comments: The analyses are well conducted and the methodology can be improved by adding the correct checklist for type of study (equatornetwork)

Response Detail comments: We thank the reviewer for this valuable suggestion. As our study is based on in vitro cellular experiments, there is no specific checklist provided by the EQUATOR Network for this research type. Nevertheless, we have revised the Materials and Methods section to enhance methodological transparency and reproducibility, in line with the EQUATOR Network’s guiding principles. We have clarified the use of biological and technical replicates, and explicitly stated the absence of randomization/blinding due to the in vitro nature of the study. A summary statement has been added at the end of the 2.10. Statistical Analysis section accordingly (see 2.10. Statistical Analysis section, lines 180–183).

Detail comments: The authors should highlight the clinical implications of this study and future perspectives in the discussion

Response Detail comments: We thank the reviewer for the valuable suggestion. In response, we have added a new paragraph at the third last paragraph of the Discussion section (see lines 570–583) to emphasize the clinical relevance of our findings and outline directions for future research.

Detail comments: How can it change the approach to the patient with CRC?, in terms of target-therapy

Response Detail comments: We thank the reviewer for raising this important clinical question. To address this, we have added a clarification in the second last paragraph Discussion section (see lines 584–597), highlighting how our findings may contribute to future targeted therapeutic strategies in colorectal cancer.

Detail comments: The introduction should be expanded by adding concepts on personalized CRC treatment:

Update on Targeted Therapy and Immunotherapy for Metastatic Colorectal Cancer. Cells. 2024 Jan 28;13(3):245. doi: 10.3390/cells13030245

Therapeutic Targets and Tumor Microenvironment in Colorectal Cancer. J Clin Med. 2021 May 25;10(11):2295. doi: 10.3390/jcm10112295

Targeted therapy for colorectal cancer metastases: A review of current methods of molecularly targeted therapy and the use of tumor biomarkers in the treatment of metastatic colorectal cancer. Cancer. 2019 Dec 1;125(23):4139-4147. doi: 10.1002/cncr.32163. Epub 2019 Aug 21. PMID: 31433498.

Response Detail comments: Thank you for your insightful suggestion. In response, we have expanded the Introduction section (the second last paragraph, lines 70–77) to include a brief overview of current advancements in personalized colorectal cancer therapy, including targeted therapy and immunotherapy. We have incorporated the key references you suggested to highlight the clinical context and unmet needs in CRC treatment, thereby reinforcing the rationale for investigating alternative therapeutic strategies such as piperine.

Detail comments: The limitation of the study must be added

Response Detail comments: Thank you for your valuable comment. In response, we have added a new paragraph at the end paragraph of the Discussion section (see lines 598–610) to acknowledge the limitations of our study, including the use of a single cell line model, the absence of in vivo validation, and the need for further pharmacokinetic and mechanistic investigations.

Reviewer 3 Report

The effects of the natural product and spice piperine on DLD-1 CRC cells was investigated in this submission. The provided experiments are accurate and the results were properly described. It is doubtful if the provided results are meaningful because of the high piperine concentrations used in this study and because only one cell line was applied for the experiments. Thus, I recommend to reconsider the manuscript after major revision:

Relatively high concentrations of piperine (62.5–250 µM) were required to achieve visible effects. It seems to be doubtful that such high concentrations can be achieved in the organism. How can piperine become an anticancer drug despite of these high effective concentrations? How can the activity of piperine be increased to reach active physiological concentrations? Please discuss.

Only one cell line was studied (DLD-1). Maybe the authors can explain and justify their choice to use only DLD-1 cells in this work. Published piperine data from other CRC cell lines might be discussed in the light of the results provided in this manuscript. Is anything known about piperine effects on MAPK and AKT signaling in other CRC cell lines (e.g., HT-29, HCT-116, etc.)?

DLD-1 has mutations in PI3K, KRAS, and p53 (see also PMID: 24042735). How far do these mutations influence the observed results? Please discuss.

Section 3.4., line 235, section 3.5., and Discussion, line 316: Please correct ´´allopurinol (a xanthine oxidase)´´, ´´allopurinol (xanthine oxidase)´´.

Figures 4 and 5: Please explain the abbreviation ´´PIP´´ in the captions of these figures.

Figure 8: Please correct ´´motochondrial´´ and the caption sentence (´´Proposed model of piperine induces apoptosis …´´).

n.a.

Author Response

Response to Reviewer’s Comments

1. Summary

We appreciate your letter notifying us about the disposition of the reviewers’ comments of our manuscript titled “Parecoxib Enhances Resveratrol against Human Colorectal Cancer Cells through Akt and TXNDC5 Inhibition and MAPK Regulation” (Manuscript ID: Antioxidants-3639424). We sincerely thank reviewers for their constructive and valuable comments that helped us improve our manuscript. Herein, we provide our point-by-point responses to the comments along with a description of all the changes that have been made and highlighted in blue in the revised manuscript.

2. Point-by-point response to Comments and Suggestions for Authors

Reviewer 3

Major comments

The effects of the natural product and spice piperine on DLD-1 CRC cells was investigated in this submission. The provided experiments are accurate and the results were properly described. It is doubtful if the provided results are meaningful because of the high piperine concentrations used in this study and because only one cell line was applied for the experiments. Thus, I recommend to reconsider the manuscript after major revision:

Relatively high concentrations of piperine (62.5–250 µM) were required to achieve visible effects. It seems to be doubtful that such high concentrations can be achieved in the organism. How can piperine become an anticancer drug despite of these high effective concentrations? How can the activity of piperine be increased to reach active physiological concentrations? Please discuss.

Response Major comments: We appreciate the reviewer’s insightful comment. In response, we have addressed this concern in the revised Discussion section (see the fourth last paragraph, Lines 550–569). It is indeed a recognized limitation that relatively high concentrations of piperine are required to elicit cytotoxic effects in vitro (62.5–250 µM). Similar effective ranges have been reported in other studies on piperine’s anticancer activity (e.g., de Almeida et al., 2020; Gusson-Zanetoni et al., 2024). However, the bioavailability of piperine is known to be relatively low in vivo due to rapid metabolism and poor solubility. To address this issue, recent research has explored various strategies to enhance the pharmacological activity and delivery of piperine, including: (1) formulation of nanoemulsions, liposomes, or nanoparticles to improve solubility and cellular uptake; (2) chemical modification or synthesis of piperine analogues with greater potency and bioavailability; (3) synergistic use with other anticancer agents to allow for dose reduction while maintaining efficacy. We have added a paragraph discussing these points and cited relevant studies in the revised manuscript. While piperine alone may not be suitable as a single anticancer drug at high doses, these enhancement strategies suggest promising directions for further development.

1. de Almeida, G.C.; Oliveira, L.F.S.; Predes, D.; Fokoue, H.H.; Kuster, R.M.; Oliveira, F.L.; Mendes, F.A.; Abreu, J.G. Piperine suppresses the Wnt/β-catenin pathway and has anti-cancer effects on colorectal cancer cells. Sci Rep. 2020 Jul 15;10(1):11681. doi: 10.1038/s41598-020-68574-2. PMID: 32669593.

2. Gusson-Zanetoni, J.P.; Cardoso, L.P.; de Sousa, S.O.; de Melo Moreira Silva, L.L.; de Oliveira Martinho, J.; Henrique, T.; Tajara, E.H.; Oliani, S.M.; Rodrigues-Lisoni, F.C. Molecular Aspects of Piperine in Signaling Pathways Associated with Inflammation in Head and Neck Cancer. Int J Mol Sci. 2024 May 25;25(11):5762. doi: 10.3390/ijms25115762. PMID: 38891950.

Major comments: Only one cell line was studied (DLD-1). Maybe the authors can explain and justify their choice to use only DLD-1 cells in this work. Published piperine data from other CRC cell lines might be discussed in the light of the results provided in this manuscript. Is anything known about piperine effects on MAPK and AKT signaling in other CRC cell lines (e.g., HT-29, HCT-116, etc.)?

Response Major comments: We thank the reviewer for this insightful comment. Initially, DLD-1 cells were selected due to their well-characterized genetic background (harboring KRASG13D, PI3KCA, and TP53 mutations), which makes them a representative model for studying oxidative stress and signaling in colorectal cancer. To strengthen the translational relevance and address this concern, we have subsequently included three additional CRC cell lines in our study: SW480, HT-29, and Caco-2. These models differ in genetic background and clinical origin, enabling us to examine the broader applicability of piperine’s anticancer effects. Our results showed that piperine consistently inhibited cell viability and induced apoptosis across all tested cell lines, suggesting that its effects are not restricted to DLD-1 cells. Regarding the reviewer’s query on MAPK and AKT signaling in other CRC cell lines (e.g., HT-29, HCT-116), we conducted an extensive literature search but found limited published data specifically linking piperine to modulation of these pathways in those models. Nevertheless, our new findings contribute novel evidence by demonstrating that piperine exerts cytotoxic effects across multiple CRC cell lines, which may potentially involve similar oxidative stress-related and signaling mechanisms. These updates have been incorporated into the revised manuscript in the Results sections (see 3.8. section, lines 332–339) and Discussion sections (see the sixth last paragraph, lines 527–540), along with the addition of the new figures (see Figure 8A and Figure 8B).

Major comments: DLD-1 has mutations in PI3K, KRAS, and p53 (see also PMID: 24042735). How far do these mutations influence the observed results? Please discuss.

Response Major comments: We appreciate the reviewer’s insightful comment. Indeed, DLD-1 cells harbor well-characterized mutations in KRASG13D, PIK3CAH1047R, and TP53, which may impact intracellular signaling dynamics. These mutations are known to activate oncogenic signaling via PI3K/Akt and MAPK pathways, which are also the major targets modulated by piperine in our study. Thus, the observed inhibitory effects of piperine on Akt phosphorylation and activation of p38/ERK signaling may partially reflect the compound’s interaction with aberrantly activated oncogenic pathways. These findings suggest that piperine may exert its anticancer activity across colorectal cancer cells through mechanisms that are at least partly mutation-independent. To address this concern, we have now included additional data using SW480, HT-29, and Caco-2 cell lines, which differ in KRAS/BRAF/TP53 status, and confirmed that piperine consistently induces cell death and PARP cleavage. This strengthens the notion that the anticancer effects of piperine are not solely confined to the specific mutational background of DLD-1 cells. We have added a new paragraph at the fifth last paragraph of the Discussion section (see lines 541–549).

Major comments: Section 3.4., line 235, section 3.5., and Discussion, line 316: Please correct ´´allopurinol (a xanthine oxidase)´´, ´´allopurinol (xanthine oxidase)´´.

Response Major comments: We have corrected to “allopurinol (a xanthine oxidase)” in the revised manuscript (see lines 238-239, line 268, line 361).

Major comments: Figures 4 and 5: Please explain the abbreviation ´´PIP´´ in the captions of these figures.

Response Major comments: PIP is the abbreviation of piperine, which we have added in the captions of these figures (see Figures 4 and 5 captions, line 251 and line 281).

Major comments: Figure 8: Please correct ´´motochondrial´´ and the caption sentence (´´Proposed model of piperine induces apoptosis …´´).

Response Major comments: The Figure 8 has been changed to Figure 9 in the revised manuscript. We thank the reviewer for identifying the typographical and grammatical errors in Figure 9. The misspelling “motochondrial” has been corrected to “mitochondrial.” The figure caption has also been revised for clarity and grammatical correctness to read: Proposed model of piperine-induced apoptosis in colorectal DLD-1 cancer cells. The corrected figure has been updated in the revised manuscript (see Figure 9 and line 624).
